# Use of the Shock Wave Therapy in Basic Research and Clinical Applications—From Bench to Bedsite

**DOI:** 10.3390/biomedicines10030568

**Published:** 2022-02-28

**Authors:** Piotr Rola, Adrian Włodarczak, Mateusz Barycki, Adrian Doroszko

**Affiliations:** 1Department of Cardiology, Provincial Specialized Hospital in Legnica, 59-220 Legnica, Poland; mateusz.barycki@gmail.com; 2Department of Cardiology, The Copper Health Centre (MCZ), 59-300 Lubin, Poland; wlodarczak.adrian@gmail.com; 3Department of Internal Medicine, Hypertension and Clinical Oncology, Wroclaw Medical University, 50-556 Wroclaw, Poland; adrian.doroszko@gmail.com

**Keywords:** shockwave, shock wave therapy, clinical application, mechanism of action, medicine, treatment technique

## Abstract

Shock Waves (SW) are acoustic disturbances that propagate through a medium carrying the energy. These specific sonic pulses are composed of two phases—high positive pressure, a rise time < 10 ns, and a tensile wave. Originally Shock Waves were introduced to clinical practice as a part of the lithotripsy therapy focused on disrupting calcific deposits in the body. Since that time, shock wave therapy (SWT) has gone far beyond the original application related to the destruction of kidney stones. In this narrative Review, we present basic clinical applications of the SWT along with the potential therapeutic application in clinical practice.

## 1. Introduction

Shock Waves (SW) are sonic pulses with unique physical properties—high positive pressure, a rise time < 10 ns, and a tensile wave. Originally, they were applied in clinical practice as lithotripsy to break up and disrupt calcific deposits in the body [1]. They were used for the first time in 1980 in Germany (Munich) for minimally invasive treatment of kidney stones. The rapid development of this technology allowed for a quick rise in the scope of performed procedures. The next step for developing lithotripsy as a therapeutic strategy was its application beyond renal diseases and implementation in the gallbladder and salivary glands diseases. After a short period of intensive development and optimizing the device-related parameters in order to minimize the risk-to-benefit ratio of such procedures, shock wave therapy has now become established as the procedure of choice for most renal calculi. Urology is not the only field where SW therapy is used in medicine. Due to numerous studies focused on Shock Waves, new therapeutic applications are being discovered and enrolled in clinical practice. This paper aims to review the most common clinical applications, including rapidly developing SW-based coronary interventions, as well as focusing on and summarizing potential fundamental molecular mechanisms responsible for the clinical effect.

## 2. Physics Basis of Shock Waves

Shock Waves are acoustic disturbances that propagate through a medium carrying the energy and consist of two phases: first, a near-instantaneous jump with coexisting small pulse width at−6dB to a high peak positive pressure, even up 100 MPa (500 bar), but more often approximately 50 to 80 MPa. After rapid rise in pressure lasting for less than 10 ns, the second phase composed of diffraction induced by tensile wave of a few microseconds duration occurs [2,3,4]. The shockwaves can occur in the broad range of frequency.

Depending on the method of application, ShockWave Therapy can be differentiated into both intra- and extracorporeal ones. The vast majority of applications used in clinical practice regard the extracorporeal SWT. There are three basic types of ESWT—focused shockwave therapy (f SWT), radial shockwave therapy (r SWT), and planar shock wave therapy (p SWT). Recently, the focused and radial SWT are most widely used [5]. Due to the level of energy supplied by SWT, we can divide it into three basic categories: low-energy, middle-energy, and high-energy shock wave therapy. There is no consensus on the borderline between these groups; however, most studies assume that the borderline between the low and middle energy is 0.1 mJ/mm^2^. The cut-off point between the middle and high energy was set up to the value of 0.28 mJ/mm^2^ [6].

There are three different types of shockwave generators used nowadays [7,8,9]. The first type is the electro-hydraulic generator, which uses the tips of an electrode as a point source. An electrical spark is generated between tips and releases a shockwave. These phenomena are associated with the vaporization of the water between the tips. The second electromagnetic generator uses an electromagnetic coil, and a metal membrane placed opposite to release shockwaves. The last one is related to a piezoelectric effect—many piezoelectric crystals are mounted in the spherical surface generator. The external energy supplies the system and forces piezoelectric crystals to contract, generating shockwaves.

As mentioned before, sound waves, along with a specific subtype, the shock wave, are nothing more than a disturbance that propagates in a medium carrying energy without permanently transporting matter; therefore, the basic physics laws ruling the behavior of sound waves in different entities define the shockwave properties. On one hand, acoustic velocity increases along with the stiffness of the material (tissue), but on the other one, it decreases with the density. When a shock wave meets a borderline between two tissues, it may be transmitted, dissipated, or reflected. In the human body, the kinetic energy carried by the wave is absorbed according to the structure of the tissues that are exposed to the shockwaves. As a rule, SW is poorly attenuated when passing through fluids and soft tissues, not causing any harm or tissue damage. While the disintegrative action focuses on hard, calcified objects, such as, for example, kidney stones [2,3,4].

The ability to fraction and disintegrate calcified stones and nodules by shockwaves is one of the basic clinical applications of SWT. Breaking stones with shockwaves is caused by several physics phenomena that may occur individually or simultaneously. First, the amplitude of the pressure of the shockwave can immediately exceed the stone pressure resistance leading to its damage because of the shock wave transition. The second phenomenon relates to a stretching wave that occurs due to reflection on the backside of the stone. This additional wave may lead to the material crushing even at low amplitude, due to the low tensile strength of mineral substances. Last is the cavitation phenomenon. On a surface of stone exposed to the wave impulses gas bubbles are being formed, but, afterwards, they collapse creating a source of additional disintegrating energy and may lead to a local increase in pressure up to 10 MPa, which results in surface erosion and additional microcracks in the stone backbone [2,3,4]. Figure 1 presents all potential mechanisms of the shockwave action leading to stone disintegration.

For the highest effectiveness of the shockwave’s degradation force, the maximum energy pulses should be focused on the point where the treatment is supposed to be administered.

The physical basis of shockwaves’ interaction with calcified structures is well explored; however, the exact impact that shock waves impart to different soft tissues and cell cultures is not fully understood. The transmission of a shock wave leads to biological effects on cell cultures and then the tissues and organs. The transformation of the physical energy into a biological response is a multi-track cascade process. In the initial phase of this process, an acoustic signal is converted by a cell into a biological reaction. Various theories [10,11] regarding how energy from shockwave therapy triggers the cellular response have been proposed. One model assumes that the mechanical deformation of cells can lead to an activation of membrane ion channels along with hyperpolarization and the modulation of membrane permeability. Another potentially involved factor is the cavitation process. Gas bubbles appear from the liquid medium due to the tensile forces of shockwaves. During implosion of the cavitation bubbles, secondary shockwaves and microjets are emitted leading to increased molecule transfection. All mentioned physical forces are transferred into a biological reaction through mechanoreceptors via mechanotransduction pathways. Mechanoreceptors are postulated to be the main conversion points of mechanical energy carried by the SW into biochemical signals affecting the activity of cells and tissues. Numerous cellular structures can play the role of mechanoreceptors–extracellular matrix proteins, stretch-activated ion channels, mitochondria, cytoskeletal components, and cell–cell junctional receptors are only a few examples [9,12]. Shockwave therapy (SWT) increases cell activity by stimulating the synthesis of various biomolecules, mainly adenosine triphosphate (ATP) [13]. The impact of SW on molecular processes is observed in various types of cells has become a cornerstone for designing clinical trials. Complete analysis of the interaction between SW and cell signal pathways is beyond the scope of this article; however, the next part of this paper summarizes the actual clinical implication of shock wave technology and their molecular basis of action. Basic clinical applications of SWT are presented in Figure 2.

## 3. Clinical Applications

### 3.1. Urology

The first implemented and well-established clinical application of ShockWave Technology was the low-invasive destruction of kidney stones [1]. In the 40 years following the invention of Extracorporeally Shock Wave Lithotripsy (ESWL) it has become the procedure of choice for most renal calculi. The mechanism of kidney stone disintegration is closely related to the physical effects of shockwaves on calcified structures and was mentioned in the previous part of this paper. Despite the long history of ESWL use, there are still some concerns about the side effects of this procedure. It has been proven that renal cells injury provoked by ESWL is not only connected with mechanical trauma [11,14,15]. The exact mechanism of this phenomenon remains unclear. Munver et al. [16] associated early renal injury after ESWL with excessive production of Reactive Oxygen Spices (ROS), but on the other hand, Sonden et al. [17] postulated it was independent from the ROS mechanism of cell destruction. In turn, both the early and late damage mechanism are strongly connected with an inflammatory response to ESWL with secondary intensification of fibrotic processes. This process is closely associated with inflammatory factors. The activation of Intercellular Adhesion Molecule1 (ICAM-1), *Tumor necrosis factor-α* (TNF-α), and *Interleukin 6* (IL-6) are postulated to play a crucial role in the early stage. While *monocyte chemoattractant protein*-*1* (MCP-1) and IL-18 may respond to the long-term renal damage [15]. Although these are not the only molecules responsible for post ESWL chronic renal impairment.

This process is related to NFκB (nuclear factor kappa-light-chain-enhancer of activated B cells) activity [18]. NFκB is a transcription factor playing an important role in the etiopathogenesis of chronic inflammatory or fibrosis diseases, due to the ability to induce transcription of fibrotic genes [19]. ESWL-related renal damage is not only connected with ROS and inflammatory process. It has also been proven that SW affects viability by activating the apoptotic cell death signaling pathway [20]. Notably, ESWL has a divergent action; it can also activate autophagy-self-protective cellular response reducing injury of renal tubular epithelial cells. Cytoprotection of autophagy seems to be dependent on the ability to regulate the Akt/GSK-3β pathway, which affects cell survival under conditions of oxidative stress.

Besides the therapy of kidney stones, shock waves a reused in the treatment of erectile dysfunction (ED) [9]. There is a strong piece of evidence suggesting the clinical effectiveness of this therapy in the group with varying severities of the disease [21]. There are many potential mechanisms responsible for this biological effect. Li et al. [22] postulated that low energy shock wave therapy can increase cell proliferation, tissue regeneration, nerve generation, and angiogenesis in a rat model of pelvic neurovascular injuries. In the process of re-innervation of penile tissue with coexisting regeneration of neuronal nitric oxide synthase (nNOS), Schwann cells are involved.

Enhanced cell activity and proliferation is achieved by the upregulation of specific markers including p-Erk1/2 and p75. Similar increasing activity of neuronal nitric oxide synthase (nNOS) was observed by Qiu et al. [23]. They postulated that SW therapy can partially resolve diabetes mellitus-associated ED by promoting nerve regeneration due to the increased synthesis of nitric oxide.

Anti-inflammatory and analgesic properties of SWT have been also applied in other fields of urology. Recent data suggest that SWT may exert a therapeutic effect in non-bacterial prostatitis. Wang et al. [24] suggest that low-energy shock wave therapy has the ability to suppress the expression of the pro-inflammatory molecules, interleukin 1β (IL-1β), cyclooxygenase- 2 (COX-2), caspase-1, Nerve growth factor (NGF), and Tumor Necrosis Factor-α (TNF-α).

A decrease in the activity of inflammatory biomarkers seems to be a potential mechanism of action in observed in improvement in symptoms in patients with interstitial cystitis/bladder pain syndrome (IC/BPS) in human [25,26]. These clinical findings have been confirmed in a recently published meta-analysis [27] regarding the therapy of patients with a chronic prostatitis/chronic pelvic pain syndrome (CP/CPPS). Additionally, similar encouraging clinical outcomes of SWT have been reported regarding the therapy of overactive bladder syndrome (OAB) [28,29]. Attenuated inflammatory responses, enhanced angiogenesis protein expression, and the elevated protein phosphorylation of ErK1/2, P38, and Akt along with areversion of mitochondria function dysregulation [30] seems to play a crucial role in these processes. Notably, the SWT can also modulate the way of administration and enhanced delivery of Botulinum toxin serotype A (BoNT/A), commonly used in the bladder and urinary tract diseases [27,31].

### 3.2. Orthopedics

The first well-proven indication of SW therapy in orthopedics was management of tendonitis. ESWs devices have been approved by the U.S. Food and Drug Administration (FDA) for the treatment of plantar fasciitis including Achilles’ tendon diseases. The effectiveness and safety of this application of SW had been confirmed in numerous randomized trials [32,33]. The mechanism of action in connective tissue diseases requires further investigation. Briefly, one of the postulated therapeutic mechanisms is the modification of the immune response—by shifting polarity in the macrophage phenotype from M1 to M2 [34]. As a general rule, the M1 population of macrophages is responsible for the pro-inflammatory response, they play an important role in the direct host-defense against pathogens such as phagocytosis and secretion of the pro-inflammatory cytokines and microbicidal molecules. On the other hand, the group of M2 macrophages limits the immune response and intensifies anti-inflammatory properties [35]. The type 2 response is known to be directly involved in regenerative processes (cell proliferation, and polyamine and collagen synthesis, releasing IL-10, and IL-4) and the promotion of angiogenesis through the release of various cytokines and growth factors. Data from animal models confirm an increase in neo-angiogenesis after shockwave therapy. Wang et al. [36,37] suggest that application of Extracorporeally Shock Wave Therapy (ESWT) caused an increasing number of neo-vessels as well as enhanced release growth and neovascularization markers including vascular endothelial growth factor (VEGF), eNOS, proliferating cell nuclear antigen (PCNA), and the *bone morphogenic protein*-2 (BMP-2).

ESWT accelerates the healing of tendon pathologies not only by modulating the immune response by vascular proliferation, but also by direct influence on human tenocytes. Vetrano et al. [38] showed up that Extracorporeal shock wave therapy promoted cell proliferation and changes in cell morphology and dedifferentiation. The authors suggest that this effect was supported by a significant increase in the levels of the Ki67 proliferation marker. Another postulated mechanism of tendinopathy was suggested by Han et al. [39] who found higher levels of matrix metalloproteinases-1, -2, and -13 (MMP-1, -2, and-13) and IL- 6 in human tendinopathy-affected tenocytes as compared with normal cells. What is interesting, the ESWT managed to reverse an unfavorable attitude and decreased the expression of several MMPs and IL-6. Enhancement of proliferation might relate to the increased level of extracellular adenosine triphosphate (ATP) after shock wave therapy, it was proven that it can trigger the release of ATP, thereby activating theErk1/2 and p38 MAPK signaling pathways.

Pain relief is also an important part of therapy in various tendinopathies. As the study suggests [40,41], ESWT might have analgesic properties. At the heart of these processes seems to involve changes in tissue concentrations of substance P and prostaglandin E2 [42]. The function of joint tendons can be improved by another kind of mechanism, ESWT can stimuli the synthesis of lubricin. This glycoprotein substance is important particularly for the tendon structures, because it facilitates tracking of the tendons. This effect can be achieved by modulation the expression of TGF-b1 [43].

Another clinical application of ESWL in the orthopedic field refers to promoting the tissue regeneration. Multiple studies have proven that shock wave therapy can be effective in reducing the time of bone healing [44,45]. Several factors have been postulated to modify this process; however, the exact mechanism is not fully understood. Wang et al. [46] claim that the clinical effect is connected to increased systematic concentration of nitro oxide level as well as change in angiogenic factors (significant elevations of VEGF, *von Willebrand factor* (vWF) and the fibroblast growth factors; furthermore, with a decrease in TGF-β) with the simultaneous growth of osteogenic factors (enhanced synthesis of BMP-2, osteocalcin, alkaline phosphatase, and the insulin-like growth factors(IGF) with co-existing lower systematic level DKK-1). Besides affecting the bone tissue, ESWT also reveals chondro protective effects. Wang et al. [37] suggest that this effect can be achieved by enhancing the synthesis of collagen II as well as higher levels of VEGF, BMP-2, and osteocalcin in the subchondral bone.

### 3.3. Cardiology

Due to numerous studies conducted on animal models and human experimental trials with the shock wave therapy, new therapeutic applications in cardiology are being discovered and enrolled in clinical practice. At the beginning, in the cardiology field, extracorporeal shock waves were tested in the porcine model of ischemic cardiomyopathy. Nishida et al. [47] have proven that SW therapy might be asafe and an effective therapeutic strategy for coronary artery disease. They showed significant improvement in recovery of left ventricular ejection fraction, wall thickening fraction, and myocardial regional blood flow. They associate the observed effects with up-regulation in the synthesis of vascular endothelial growth factor (VEGF). A similar effect in the prevention of early ischemic left ventricular remodeling by SWT was observed by Uwatoku et al. [48]. The exact mechanism of these changes remains unclear. Most of the available studies point to the influence on angiogenesis as a possible mechanism of action. Tepekoylu et al. [49] suggested that increased vascular formation might be mediated via RNA/protein complexes with involvement of the antimicrobial peptide LL37 activating TLR3 (Toll-like receptor 3). A completely different mode of healing action of SWT in human heart failure is presented by Assmus et al. [50] They examined the effect of intracoronary administration of autologous bone marrow-derived mononuclear cells in subjects with post-infarction chronic heart failure (HFrEF, heart failure with reduced ejection fraction). Additionally, one of the groups received the subsequent application of SWT. As a result, post shock wave group-facilitated intracoronary administration of bone marrow-derived mononuclear cells led to a significant improvement in the left ventricular ejection fraction. Authors postulated a connection between the extracorporeal application of low-energy shock waves and increased expression of chemoattractant such as stromal cell-derived factor 1 (SCDF-1) and VEGF in the target tissues. Notably, the authors claim that pure shock therapy (without administration of autologous bone marrow-derived mononuclear cells) did not result in any improvement in cardiac muscle function.

There are few clinical studies focused on the influence of SWT on patients with chronic refractory angina [51,52,53]. Vainer et al. [52] proven that cardiac shockwave therapy improved symptoms and reduced ischemia burden in patients with end-stage coronary artery disease. They investigated the clinical response of 33 humans on the 4-month outpatient shockwave treatment protocol. It was found that SWT reduced the use of sublingual nitrates and decreased angina complaints (at least one CCS class). What was most encouraging for further study was that they noticed a significant decrease in ischemia burden in the treated regions, as assessed using SPECT stress images. This might suggest that at least a part of the positive effect is correlated with improved perfusion in the treated ischemic zones. More details regarding the molecular mechanism of this phenomenon are highlighted in a paper by Chai et al. [54], which postulates that one of the mechanisms involved in the SWT anti-ischemic model of action could rely on neovascularization and increase the number and activation of circulating endothelial progenitor cells. The authors suggest that higher levels of IL-8 and VEGF are mediators of these processes.

ShockWave technology is not only dedicated to patients with severe coronary artery disease (CAD) without revascularization options. Despite significant developments in percutaneous coronary intervention techniques, heavy calcification in arterial lumen remains one of the greatest challenges in the treatment of coronary artery disease. Recently, a novel technology—The Shockwave Medical Intravascular Lithotripsy System (Shockwave Medical Inc., Fremont, California, USA) has been proposed as a resolution for this clinical problem, which is based on the “classical” lithotripsy model of action. The impulse-generating balloon catheter-device is placed in the body of calcified atherosclerotic lesion, through regular angioplasty guidewire (intra-coronary), and releases Shock Wave pulses. The effectiveness of this solution in clinical practice has been proven, both for peripheral and coronary artery disease [55,56,57], as shown in our previous clinical studies. Even though, this technology appears to be safe, there are a few concerns about potential side-effects—especially connected with proarrhythmic potential [58]. Due to the observed activation of platelets [59] resulting directly from the shockwave therapy, concerns about an increased tendency to platelet aggregation are reasonable. Nevertheless, as far as the literature is concerned, no studies have been performed on either in vitro or in vivo models to solve this problem. Available data suggesting an increase in platelet activation under shock waves, only reveal an increase in the release of multiple growth factors and cytokines from their α granules including TGF-β1 and PDGF-ββ [59]. In coronary artery disease, attempts are also being made to use direct cardiac shock wave therapy, during surgical revascularization-coronary artery bypass grafting [60,61]. Unfortunately, results in humans have not been published yet but results of animal models are promising [62].

### 3.4. Dermatology

Among all the clinical applications of the shock wave in dermatology, the ability to enhance tissues regeneration is most often applied. Several studies have assessed the effectiveness and safety of shockwave therapy in the treatment of patients with chronic leg ulcers [63,64,65,66,67]. Chronic foot ulcers are defined as non-healing ulcers with a duration time of at least 3months. The etiology of such ulcers is diversified. Conventionally they can be divided into two main groups: diabetic and non-diabetic. Non-diabetic ulcers are caused by peripheral arterial disease, post-traumatic skin lesions, infections, deep vein thrombosis, or venous stasis with poor venous return. Managements of chronic foot ulcers require multidisciplinary approaches such as wound care, surgery, antibiotics, diabetic control, or compression therapy [65]. However, sometimes it is insufficient, and alternative therapy such as oxygen, larvae, or laser radiation therapy is needed. Wang et al. [66] showed that extracorporeal shockwave therapy (ESWT) is effective and safe in the treatment of chronic foot ulcers in short- and long-term therapy. They reported better clinical results in the non-diabetic group than the opposite one. Despite all the differences, both groups achieve long-term improvement. The authors combine the positive results with significant improvement in blood flow perfusion rate in the treated skin area. The exact mechanism of ESWT remains unclear but data from animal experiments [38,39,66,68] suggest that SWT could up-regulate angiogenic and osteogenic growth factors: endothelial nitric oxide synthase, vascular endothelial growth factor, bone morphogenetic protein 2, and proliferating cell nuclear antigen. 

Not only ulcers are in the spotlight of Shockwave therapy in dermatology. Schaden et al. [69] tested the use of unfocused, low-energy ESWT on a large population of patients with acute or chronic soft tissue wounds. This registry indicates that shock wave therapy is safe and effective in “real life, everyday practice”. They postulated several benefits of SWT including ease of application, minimally invasive, low-profile side-effect (including infection, or drug interaction issue), and painlessness (patients do not require additional anesthetics).

Hypertrophic and contracture scars after burn injuries can cause functional and cosmetic deformities. Fioramonti et al. [70] observed improvement in scar appearance after application of SW in postburn pathologic scars. The explanation for the clinical benefits of ESWT is probably changing the expression of fibrosis-related molecules in fibroblasts. To examine these phenomena, Cui et al. [71] derived fibroblasts from human hypertrophic scars and investigated the influent of shockwave therapy on them. They demonstrated that ESWT did not have any impact on the viability of fibroblasts, but it decreased migration as well as inhibited expression of transforming growth factor-beta 1 (TGF-β1), alpha-smooth muscle actin (α-SMA), collagen-I, fibronectin, and twist-1. They noticed increased expression of E-cadherin with coexisting reduction in N-cadherin. In conclusion, they put forward a proposal that suppressed epithelial–mesenchymal transition might be responsible for the anti-scarring effect of ESWT and is a potential therapeutic target in the management of post-burn scars.

### 3.5. Neurology

Stroke is, now a days, among the leading causes of disability. Numerous patients who experienced stroke have decreased health-related quality of life, mainly due to the presence of spasticity. Despite various proposed treatments (invasive and non-invasive), spasticity remains a burning clinical problem. A growing amount of evidence [72,73] suggests the usefulness of ESWT in neurological conditions. Particularly in the therapy of post-stroke muscle spasticity [74]. Furthermore, shockwave therapy is effective alone and as a support of conventional physiotherapy or botulinum toxin. Studies conducted so far suggest that ESWT affects the rheological properties of the spastic muscle. The mechanism of action is based on intrinsic hypertonia or spasticity (extracellular-matrix and muscle fibrosis). Notably, no major complications or side effects occur after the treatment, as reported in the previously mentioned meta-analysis.

Since conventional treatments for postherpetic neuralgia (PHN) and postherpetic itch (PHI) are insufficient, novel therapeutic methods are urgently needed. Lee et al. [75] postulated that ESW has the ability to decrease significantly PHN and PHI. Unfortunately, the exact mechanism of this process remains unclear.

Post-herpetic neuralgia is not the only neurological condition where analgesic properties of ESW are used. Low back pain (LBP) is one of the leading causes of chronic pain worldwide, generating significant costs for the healthcare system. Walewicz et al. [76] conducted a randomized clinical trial, which suggests that ESWT can have a long-term impact on reducing chronic low back pain and may lead to significant improvement in the postural sway in patients with LBP compared with standard core stability training.

Since Shock Wave Therapy shows the ability to enhance metabolic activity and increase the proliferation potential of various cells, regenerative properties of SWT have been used to promote peripheral nerve regeneration [77,78]. Li et al. [78] showed a post-SWT significant promotion of axonal regrowth and increased myelination expression, where the potential molecular mechanism is related to increased expression of two mechanosensitive transcription factors YAP/TAZ proteins. Regenerative and neuroprotection properties of SWT have been applied in spinal cord injury models. Lobenwein et al. [79] along with Gollmann-Tepeköylü [80] indicate that Toll-like receptor (TLR) 3 signaling is involved in neuroprotection and spinal cord repair. Additionally, vascular endothelial growth factor (VEGF) [81] related mechanisms are postulated to be involved in this process. Moreover, since SWT can activate stem cells [82,83,84], along with the ongoing development of this treatment method, a potentially wide range of applications in the different fields of regenerative medicine can be applied [85,86,87,88]—not only those related to the central or peripheral nervous system.

Interestingly, some data support the use of SWT in structural spinal diseases—scoliosis. Several reports [89] have postulated lower spinal cord deformity and deviation of the spinal cord in the central can alin humans.

All clinical application along with mechanism of action of Shockwave Therapy are summarized in Figure 3.

## 4. Future Perspectives

The field of clinical applications of Shockwave Therapy is constantly expanding, and new clinical trials focused on an investigation of novel therapeutic applications are currently ongoing. In addition to the previously mentioned well-established clinical applications—several currently ongoing trials aim to evaluate the safety and efficiency of SWT, including patients with moderate kidney failure (NCT02515461), undergoing surgical treatment of coronary artery disease (NCT03859466), rehabilitation after prostatectomy (NCT03862599), recovery after chemotherapy (NCT05111730), and therapy of a urinary tract tumors (NCT04644835). Results of these trials along with outcomes of the basic research studies may open a completely new perspective for SWT. Although, no severe complications of the SWT appliances have been described so far, the safety concerns are not unfounded. On one hand, it has been proven that SWT can modulate the proliferation [90] and destruction of cells as well as modulate the immune response [64]. On the other hand, SWT is commonly used as a part of the rehabilitation and a recovery process in patients with malignancies [91]. Future studies focused on the influence of SWT on different tumor biology and potentially carcinogenic mechanisms of action are needed before the wide application of this therapy in oncological patients.

## 5. Conclusions

Since the date of invention, shock wave therapy has gone far beyond the original application related to the destruction of kidney stones. Beyond basic research, it has been widely applied in many fields of clinical practice. Thanks to continuous development of the technology and the falling expenses of therapy in the future, Shockwave Therapy might become a useful complementary tool in the treatment of numerous chronic disorders. 

Nevertheless, to verify the effectiveness and safety of this therapeutic approach, further clinical studies aimed at the explanation of its therapeutic mechanisms are required. Even though the side effects of SWT are hardly observable, future studies focusing on this issue are needed, and special attention should be paid to the safety of intracorporal Shock Wave Therapy, such as during percutaneous coronary interventions using shockwave coronary lithotripsy (SIVL-PCI).

## Figures and Tables

**Figure 1 biomedicines-10-00568-f001:**
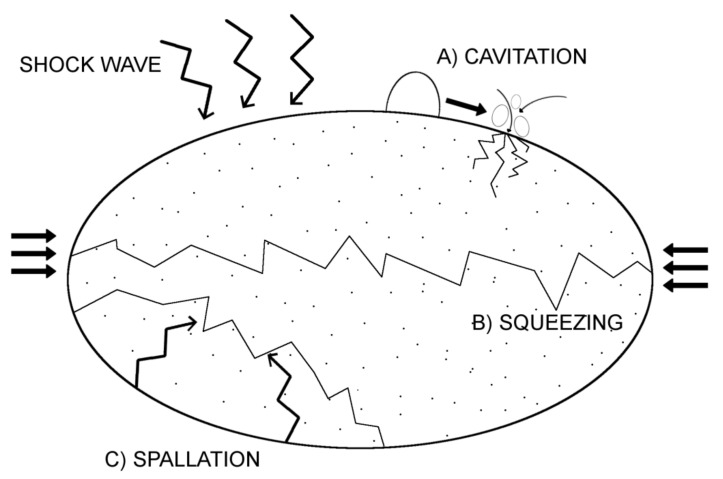
Mechanism of Shock Wave action-related stone disintegration.

**Figure 2 biomedicines-10-00568-f002:**
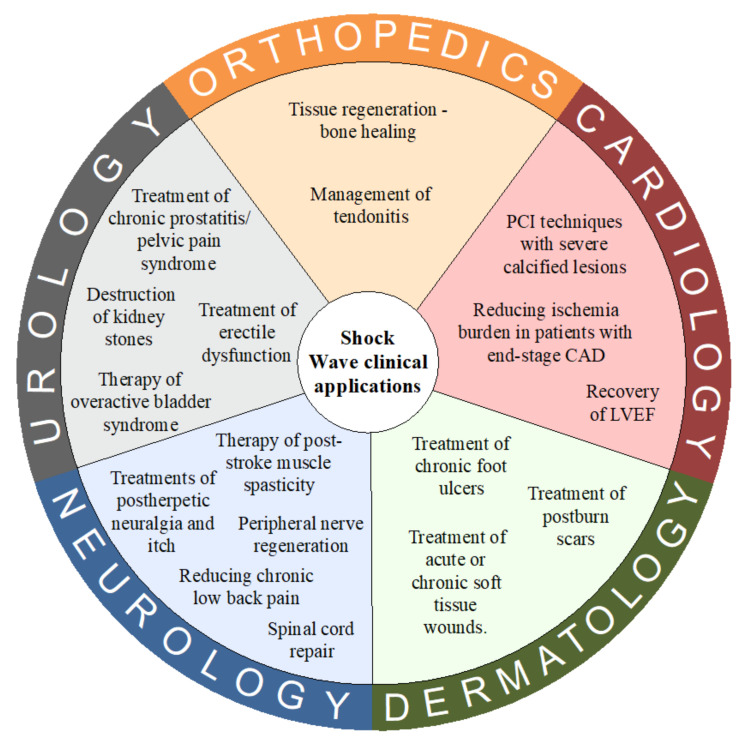
Basic clinical applications of Shock Wave Therapy. Abbreviations: CAD = coronary artery diseases; LVEF = left ventricle ejection fraction.

**Figure 3 biomedicines-10-00568-f003:**
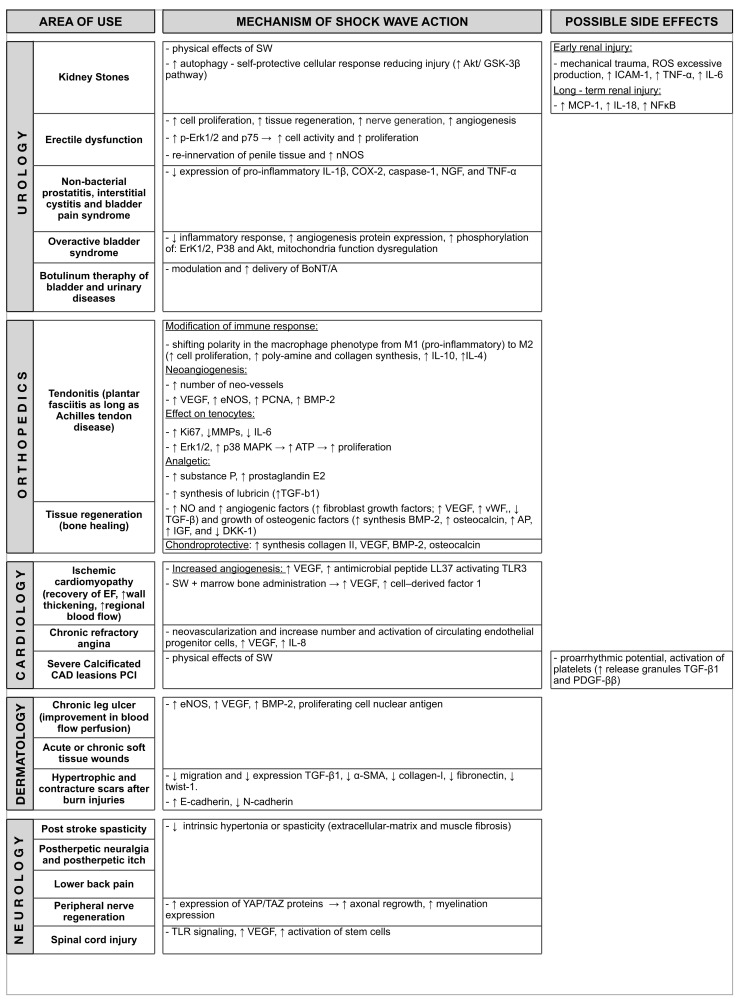
Clinical application and mechanism of action of Shock Wave Therapy. Abbreviations: SW = shock wave; ICAM1 = Intracellular Adhesion Molecule1; TNF-α = Tumor necrosis factor-α; IL-6 = Interleukin 6; ROS = Reactive Oxygen Species; NFκB = nuclear factor kappa-light-chain-enhancer of activated B; NOS = nitric oxide synthase; ATP = adenosine triphosphate; TGF = transforming growth factors; MMP = matrix metalloproteinase; VEGF = vascular endothelial growth factor; SCDF = stromal cell-derived factor; TLR = Toll-like receptor; SMA = smooth muscle actin; vWF = Willebrand factor; BMP = bone morphogenic protein; IGF = insulin-like growth factor; NO = Nitric Oxide; MCP = monocyte chemoattractant protein.

## Data Availability

Not applicable.

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
