# Peer review of "Use of the Shock Wave Therapy in Basic Research and Clinical Applications—From Bench to Bedsite"

_biomedicines, 2022, doi:10.3390/biomedicines10030568_

Round 1

Reviewer 1 Report

In this review manuscript, the authors summarized the clinical application of shock wave therapy in urology, orthopedics, cardiology, and neurology. The mechanisms of applications were well presented. Please see below for minor comments:

  1. Are there any ongoing clinical trials?
  2. Please further discuss the potential limitations of clinical application.

Author Response

We would like to thank the Reviewer for an in-depth analysis of the manuscript and for pivotal comments provided, which have resulted in a significant improvement of this manuscript.

AD 1 and 2 . As the Reviewer suggested, we provided a comment on future perspective of SWT with short summarize of ongoing trials along we discussed  the potential limitations of clinical application of SWT- “  The field of clinical application of Shock Wave Therapy is constantly expanding, and new clinical trials focused on an investigation of novel therapeutic applications of SWT are conducted every day. Beyond mentioned - well-established clinical applications- several currently ongoing trials aim to evaluate the safety and efficiency of SWT, including patients with moderate kidney failure (NCT02515461), undergoing surgical treatment of coronary artery disease (NCT03859466), rehabilitation after prostatectomy (NCT03862599), recovery after chemotherapy (NCT05111730), therapy of a urinary tract tumors (NCT04644835). Results of these trials along with outcomes of the basic research studies may open a completely new perspective for SWT. Although, no severe complications of appliances of SWT have been described so far, the safety concerns are not unfounded. On one hand, it has been proven that SWT can modulate the proliferation [67] and destruction of cells as well as modulate the immune response[68]. On the other hand, SWT is commonly used as a part of the rehabilitation and a recovery process in patients with malignancies[69]. Future studies focused on the influence of SWT on different tumors biology, and potentially carcinogenic mechanisms of action are needed before wide applications of this therapy in oncological patients.

Reviewer 2 Report

This manuscript reviews the most common clinical applications of shock waves therapy, starting from the more commun to the more recent ones.  The manuscript is fluently written and understandable, even if several typos as well as some English errors are evident through the manuscript.  Authors referred to the advantages of this technique and its added value is the  explanation of the mechanisms of the clinical activity. 

The strength is mainly in the subject of the review that refers to a technique largely implemented during the last decade, with further potential and in expansion.  There are no specific weaknesses. The addition of some figures helping the reader in understanding the different applications and their mechanism could be useful. Try to add more references if possible. 

Author Response

We would like to thank the Reviewer for an in-depth analysis of the manuscript and for pivotal comments provided, which have resulted in a significant improvement of this manuscript.

As the Reviewer suggested, we provided additional Figures to the body of the manuscript to improve the clarity of the paper and achieve proper illustration to the readers' SWT clinical applications.

Another valid point mentioned by the Reviewer is a remark regarding increasing the number of references used. We revised the manuscript and extended references database.
